# Recent Advances in Diagnosis and Treatment Approaches in Fungal Keratitis: A Narrative Review

**DOI:** 10.3390/microorganisms12010161

**Published:** 2024-01-13

**Authors:** Laura Andreea Ghenciu, Alexandra Corina Faur, Sorin Lucian Bolintineanu, Madalina Casiana Salavat, Anca Laura Maghiari

**Affiliations:** 1Department III Functional Sciences, Victor Babes University of Medicine and Pharmacy, E. Murgu Sq., no. 2, 300041 Timisoara, Romania; bolintineanu.laura@umft.ro; 2Department IX Surgery, Discipline of Ophtalmology, Victor Babes University of Medicine and Pharmacy, E. Murgu Sq., no. 2, 300041 Timisoara, Romania; palfi.madalina@umft.ro; 3Department of Anatomy and Embryology, Victor Babes University of Medicine and Pharmacy, E. Murgu Sq., no. 2, 300041 Timisoara, Romania; s.bolintineanu@umft.ro (S.L.B.); boscu.anca@umft.ro (A.L.M.)

**Keywords:** fungal keratitis, mycotic keratitis, culture growth, antifungal therapy, diagnostic approach

## Abstract

Fungal keratitis represents a potentially sight-threatening infection associated with poor prognosis, as well as financial burden. Novel diagnostic methods include polymerase-chain-reaction (PCR)-based approaches, metagenomic deep sequences, in vivo confocal microscopy, and antifungal susceptibility testing. The ideal therapeutic approaches and outcomes have been widely discussed in recent times, with early therapy being of the utmost importance for the preservation of visual acuity, minimizing corneal damage and reducing the scar size. However, combination therapy can be more efficacious compared to monotherapy. Understanding the pathogenesis, early diagnosis, and prevention strategies can be of great importance. In this narrative, we discuss the recent progress that may aid our understanding of the diagnosis, treatment, and prevention of mycotic keratitis.

## 1. Introduction

Fungal keratitis is a severe corneal infection characterized by the invasion by fungal organisms of the corneal tissue. The condition primarily affects individuals with compromised corneal integrity, often due to trauma, corneal abrasions, or the use of contact lenses. Among 105 known pathogenic mycotic species of keratitis, *Aspergillus*, *Fusarium*, and *Candida* are the most common and are accountable for 70% of the keratitis-causing fungal species [1].

Mycotic keratitis stands as a leading cause of ocular morbidity worldwide, with a higher incidence in tropical and subtropical regions [2,3]. A 2001 survey by the World Health Organization revealed that corneal blindness ranks as the second-leading cause of blindness, following cataracts [1]. Particularly in developing countries, ocular trauma and corneal ulceration are known as critical factors leading to corneal blindness. 

Clinical manifestations of fungal keratitis include intense ocular pain, conjunctival hyperemia, photophobia, blurred vision, and epiphora. The infection may progress to the formation of corneal ulcers, leading to necrosis and potential scarring. Timely and accurate diagnosis is essential for appropriate management, typically involving a comprehensive eye examination, corneal scrapings for microbiological analysis, and the identification of the specific fungal pathogen [4].

The treatment of fungal keratitis involves a multifaceted approach aimed at eradicating the fungal infection, minimizing corneal damage, and preserving visual function. The choice of treatment depends on factors such as the severity of the infection, the specific fungal pathogen involved, and the response to initial therapeutic measures. Topical antifungal medications such as natamycin and voriconazole are frequently used. Voriconazole, in particular, has shown efficacy against a broad spectrum of fungi and is often considered a first-line treatment [5]. Amphotericin B is another potent antifungal agent which may be used topically. However, its use is sometimes limited by concerns about corneal toxicity. In severe cases, or when the infection is not adequately controlled with topical medications alone, systemic antifungal therapy may be considered. Oral voriconazole or itraconazole is commonly prescribed, especially when there is a risk of systemic dissemination. In cases of advanced or refractory infections, surgical interventions such as corneal debridement or, in extreme cases, therapeutic penetrating keratoplasty (PK) may be necessary to control the spread of the infection and salvage visual function [4,6]. 

Fungal keratitis often requires a collaborative effort between ophthalmologists, infectious disease specialists, and microbiologists. Regular follow-up visits are essential to assess the effectiveness of treatment and address any complications promptly. Early diagnosis, aggressive treatment, and close monitoring are critical components in achieving optimal therapeutic outcomes and preventing potential complications such as corneal scarring and visual impairment [7].

Preventative measures focus on meticulous eye care, particularly for contact lens wearers. Adherence to proper hygiene practices, regular lens cleaning, and avoiding overnight wear can mitigate the risk of fungal keratitis. Additionally, prompt attention to corneal injuries and the avoidance of environmental factors that may predispose individuals to fungal infections contribute to preventive efforts. Despite advancements in diagnostic and therapeutic modalities, the management of fungal keratitis remains intricate, requiring a multidisciplinary approach for optimal outcomes [2].

## 2. Fungal Diversity in Keratitis

Fungi represent a vast and diverse kingdom with various ecological roles and impacts on human health [8,9]. *Fusarium*, a genus of filamentous fungi, encompasses species that can impact human health both directly and indirectly. While primarily recognized for causing plant diseases, some *Fusarium* species pose a threat to human health, particularly in individuals with weakened immune systems. Fusarium can cause serious infections in humans, including ocular, skin, and nail infections, and more critically, invasive diseases such as bloodstream infections, especially in immunocompromised individuals or those undergoing invasive medical procedures. Moreover, certain *Fusarium* species produce mycotoxins, such as fumonisins and trichothecenes, which can contaminate food crops. Ingestion of these mycotoxins has been associated with various health issues, making *Fusarium* a concern in both clinical and agricultural contexts [10].

*Aspergillus*, a diverse type of fungus, includes species that commonly interact with humans. *Aspergillus fumigatus*, in particular, is a leading cause of invasive aspergillosis, a serious respiratory infection affecting immunocompromised individuals. Aspergillosis can manifest as lung infections, allergic reactions, or invasive diseases with high mortality rates [11]. Beyond its clinical significance, *Aspergillus* species contribute to human activities positively and negatively. Some species, like *Aspergillus oryzae*, are crucial in food fermentation processes, while others, such as *Aspergillus flavus*, produce aflatoxins that contaminate crops and pose health risks. Aflatoxins are potent carcinogens, and their presence in food, especially in regions with inadequate storage conditions, raises concerns for human health [12].

*Candida*, a type of yeast-like fungi, is intimately associated with the human body as part of its natural microbial community. *Candida albicans*, the most prevalent species, typically resides in the gastrointestinal and genitourinary tracts. Under normal conditions, *Candida* is a commensal organism. However, factors like antibiotic use, immunosuppression, or hormonal changes can disrupt the balance, leading to *Candida* overgrowth and causing candidiasis [13]. 

Apart from *Fusarium*, *Candida*, and *Aspergillus*, other types of fungi can also contribute to fungal keratitis. Certain molds and yeasts, such as *Alternaria*, *Curvularia*, and *Bipolaris*, have been implicated in fungal keratitis cases. These fungi are often found in environmental sources, including soil and plant materials [14].

## 3. Diagnosis of Fungal Keratitis

Filamentous fungal keratitis typically manifests in young men involved in agricultural or outdoor activities. The fungi responsible for these infections do not breach an intact epithelium spontaneously; rather, invasion commonly occurs as a secondary consequence of trauma. Trauma stands out as the primary predisposing factor, affecting 40–60% of patients [15,16]. Other identified risk factors are prior ocular surgery, ocular surface disease, previous use of corticosteroids (topical or systemic), and contact lens usage [17,18,19]. Notably, a study on mycotic keratitis demonstrated the following: antifungal therapy or surgical intervention led to no response in patients with previous ocular surgery, a partial response in those with ocular trauma, a notable response in individuals with ocular surface disease, a universal response in contact lens users, and varied responses in patients with a history of corticosteroid use [5].

Filamentous fungal keratitis is predominantly caused by species like *Fusarium*, *Aspergillus*, *Curvularia*, and other phaeohyphomycetes. Environmental factors such as humidity, rainfall, and wind seem to influence the occurrence of filamentous fungal keratitis, contributing to seasonal variations in fungal isolation frequency and the specific fungal species identified [19,20]. 

Keratitis resulting from infection with *Candida albicans* and other related fungi is often associated with one or more ocular conditions, such as inadequate tear secretion or impaired eyelid closure, as well as systemic factors like diabetes mellitus or immunosuppression, which can predispose individuals to such infections. Additionally, this type of mycotic keratitis may develop in the presence of a pre-existing epithelial defect, either due to herpes keratitis or caused by abrasions resulting from contaminated contact lenses. The interplay of these local and systemic factors contributes to the susceptibility and occurrence of Candida-related keratitis [15].

The laboratory diagnosis of fungal keratitis has an important significance in facilitating appropriate and effective treatment. It has a role in conducting antifungal susceptibility testing to determine the responsiveness of patients to both traditional and newer antifungal agents. Typically, specimens for microbiological evaluation are obtained through corneal scrapings, while cases with deeper infiltrates or those resistant to standard procedures may necessitate corneal biopsy or suture biopsy. Traditional diagnostic approaches for fungal keratitis involve staining of the smear and culturing of corneal scrapings. However, contemporary molecular diagnostic methods, such as polymerase chain reaction (PCR), are nowadays valued due to their precision and rapidity. Additionally, real-time in vivo confocal microscopy (IVCM) corneal imaging is emerging as a valuable tool for early detection by identifying fungal hyphae in cases of fungal keratitis [2,4]. A brief summary of diagnostic methods is shown in Table 1 [2,10,21,22,23,24,25].

### 3.1. Microscopic Examination

The conventional diagnostic approach for keratitis involves the microscopic examination of corneal scraping specimens to detect fungal elements. Various stains are used for this purpose, including Gram, 10% potassium hydroxide (KOH) wet mount, calcofluor white, lactophenol cotton blue, Giemsa, acridine orange, and Periodic Acid Schiff (PAS) stains [16]. The identification of fungal hyphae using KOH is widely utilized for provisional diagnoses in many regions due to its cost effectiveness, straightforward procedure, ready availability, and ability to yield rapid results, facilitating a rapid start to antifungal therapy. The sensitivity and specificity of preparations involving KOH have been reported in the specialized literature as between 60% and 99.3% [26,27], and between 70% and 99.1% [17,20].

The effectiveness of staining methods varies depending on factors such as the specific stain employed, the skill of the medical doctor, and the nature and amount of the sample. When conventional techniques like corneal smear staining and culture yield no identifiable organisms, when the disease continues to progress in spite of maximum treatment, or when corneal involvement is too profound for scraping, resorting to a corneal biopsy becomes imperative. The use of specialized stains can improve the visibility of fungal hyphae and yeast. Several studies suggest that corneal biopsy specimens may exhibit greater sensitivity compared to scraping samples, possibly attributed to factors like deep stromal engagement by certain fungi or the restricted amount of corneal material obtained through scraping [28].

The microscopic examination of corneal scrapings in fungal keratitis aids in rapid and cost-effective presumptive diagnosis and leads to the direct visualization of fungal structures. Therefore, this method has often been used for the prompt initiation of targeted antifungal therapy, which can be crucial in preventing progression. Moreover, in developing countries, microscopic examination can be the only available diagnostic tool.

### 3.2. Culture Growth

While time-consuming, this method is indispensable for species identification and effective treatment. Additionally, it enables antifungal susceptibility testing to determine sensitivity to both traditional and newer antifungal agents. Culturing is considered the gold standard in fungal keratitis diagnosis due to its high specificity. Commonly employed culture media include Sabouraud’s dextrose/potato dextrose/blood/chocolate/thioglycolate agar. Despite its reliability, culture results may take over a week to exhibit results, potentially delaying the diagnosis. The corneal material has to be spread out as thinly as possible on the slides in order to facilitate visualization of the fungal hyphae or yeast cells. Other limitations include low sensitivity rates, the necessity for an experienced microbiologist for result interpretation, and the challenge of distinguishing between species that exhibit morphologically similar growth. These considerations show the need for a comprehensive approach that incorporates various diagnostic methods to overcome the limitations associated with culture alone [26,29].

The failure of traditional methods like clinical evaluation, corneal scrapings, and initial culture can result in suboptimal management, leading to failure in healing and visual impairment. A novel approach involves the endorsement of repeat cultures as a diagnostic and prognostic tool after initiating the treatment. A secondary analysis of data from the mycotic ulcer treatment trial (MUTT)-1 (milder, smaller ulcers) [22] and MUTT-2 (severe ulcers) [30] demonstrated that positive repeat cultures performed six days after treatment initiation were linked to impairment of visual acuity at the three-month mark, larger scar dimensions, and an elevated incidence of perforation and/or the requirement for therapeutic PK. Consequently, a repeat culture at day six serves as a crucial prognostic indicator, signaling the need for close monitoring, potential adjustments to the treatment, and consideration of early surgical interventions like PK or lamellar procedures such as therapeutic deep anterior lamellar keratoplasty (DALK) in positive cases [6,31]. These repeat cultures play an important role in evaluating the effectiveness of both conventional and newer antifungal agents. Consequently, sixth-day cultures are now recommended as a crucial prognostic tool, and ongoing research in this direction is likely to further establish their significance [32].

While culture is considered the gold standard for the diagnosis of mycotic keratitis, its main disadvantage is that it may take several days to weeks to receive a result and, therefore, it can withhold prompt treatment, which is crucial in fungal keratitis.

### 3.3. In Vivo Confocal Microscopy

IVCM represents an innovative and noninvasive technique used for analyzing the cornea, using a series of pinhole apertures to create optical sections. This technology shows each corneal layer, similar to in vitro histochemical techniques. Fungal keratitis, characterized by nonspecific clinical features, poor yields on scraping specimens, variable sensitivity of culture results, and the extended time required for culture growth, often leads to delayed diagnoses and suboptimal treatment outcomes. IVCM addresses these challenges and offers the additional benefit of being noninvasive [21]. In addition to diagnosis, IVCM may also be used to monitor the response of fungal keratitis to treatment. After 1 month of antifungal therapy to a patient with infection with *Alternaria alternata*, IVCM demonstrated a significant reduction in inflammatory cells and showed the presence of hyper-reflective scar-like tissue and the absence of branching hyphal infiltrates in the affected cornea [2].

Studies on IVCM in infectious keratitis have demonstrated promising results. Kanavi used tandem scanning–IVCM and reported high sensitivity and specificity percentages for mycotic keratitis (94% and 78%, respectively) [33]. In a study by Chidambaram et al., a laser-scanning confocal microscope achieved a sensitivity of 85.7% and a specificity of 81.4% in detecting fungal filaments [34]. A few advantages include being a noninvasive technique and the early identification of fungi, as well as the monitoring and guidance of treatment. IVCM also has limitations, including being a contact procedure, requiring a cooperative patient in the symptomatic stage, higher expenses, and restricted availability, and the incapacity to identify organisms at the species level currently limits its application as a primary diagnostic approach.

IVCM is a noninvasive technology that leads to the direct observation of fungal elements, such as hyphae and spores, and aids in the diagnosis, management, and follow-up of cases with mycotic keratitis. Although very promising, it still has its disadvantages, such as the need for an expert operator and patient cooperation.

### 3.4. Antifungal Susceptibility Testing

As antifungal resistance increases and new antifungal agents are introduced, antifungal susceptibility testing (AFST) and minimum inhibitory concentration (MIC) determination play a crucial part in the effective treatment of mycotic keratitis. The primary objective of AFST is to provide important information for clinicians regarding the susceptibility before and during treatment, or a resistance phenotype related to a particular combination of organism and antifungal agent. While the treatment of choice for various types of fungi can be empirically assumed based on proper pathogen identification, susceptibility testing becomes particularly useful when invasive mycotic infections are present, when developed drug resistance is suspected, or in patients unresponsive to treatment [2].

Two globally acknowledged committees, the Clinical and Laboratory Standards Institute (CLSI) and the European Committee for Antimicrobial Susceptibility Testing (EUCAST), have established phenotypic assays for in vitro AFST based on the broth dilution method for *Aspergillus* and *Candida* species [35,36,37]. The MIC is the lowest concentration of an antimycotic agent that inhibits the visible growth of a microorganism after a defined period of incubation. It is a key parameter in AFST, providing information about the effectiveness of an agent against a specific fungi. CLSI has provided protocols on MIC values for yeasts, while EUCAST has set MIC values for various antifungal agents against specific *Aspergillus* and *Candida* species. However, comprehensive information regarding MIC breakpoints for other species is still needed. 

AFST, performed in clinical microbiology laboratories to assist in selecting the appropriate treatment, has demonstrated a connection between susceptibility and the response to treatment [38,39]. In a recent study that determined the MICs of natamycin and voriconazole on isolates from mycotic keratitis, Lalitha et al. demonstrated that natamycin had greater breakpoint values against all specimens except for *Fusarium* spp., while voriconazole had the lowest breakpoint value targeting *Aspergillus* species. They also showed that the greater the MIC breakpoint value, the greater the odds of developing corneal perforation [40]. Saha et al. assessed AFST using the disk diffusion method. They observed that *Aspergillus* spp. and *Fusarium* sp. exhibited higher sensitivity to voriconazole than natamycin, while amphotericin B showed effectiveness against yeasts [40]. Patil showed variable MIC against Candida with a range from 1–2 μg/mL for *C. albicans* [41], while Salvosa reported that MIC can be as high as 150 μg/mL for *C. parasilopsis* [42].

AFST plays a pivotal role in guiding the effective management of fungal infections and is integral to optimizing patient outcomes. Utilizing AFST allows clinicians to identify the most appropriate antifungal therapy tailored to the specific susceptibility profile of the infecting strain.

### 3.5. Molecular Diagnostic Techniques

#### 3.5.1. Molecular Diagnostic Techniques Applied to Isolates Derived from Cultures

Molecular diagnostic techniques have revolutionized the rapid diagnosis of fungal keratitis, using PCR-based approaches. They play an important role in enhancing the accuracy and efficiency of identifying cultured organisms in cases of fungal keratitis. PCR is a commonly employed molecular method that amplifies specific DNA sequences, allowing for the rapid and sensitive detection of fungal pathogens. PCR-based assays, such as species-specific PCR and multiplex PCR, enable the differentiation of various fungal species and strains directly from cultured samples. DNA sequencing is another powerful technique, providing detailed information about the genetic makeup of the isolated organisms. Sequencing methods, including Sanger sequencing and next-generation sequencing (NGS), facilitate the identification of fungi at the species level, even in cases of complex and mixed infections [43]. Additionally, real-time quantitative PCR (qPCR) allows for the quantification of fungal DNA, aiding in assessing the severity of the infection [44]. These molecular techniques not only streamline the identification process but also contribute to a deeper understanding of the genetic diversity and epidemiology of fungal keratitis, guiding clinicians in tailoring appropriate antifungal treatments.

Matrix-assisted laser desorption ionization time-of-flight mass spectrometry (MALDI-TOF MS) is a method used for identifying pathogens within minutes [45]. Initially intended for bacterial organisms, it is now thought to be a tool for identifying fungal isolates, particularly yeasts and some genera of filamentous fungi [46]. A study found that this technique, used to detect the etiological spectrum of infectious keratitis, accurately identified pathological microorganisms in 51%, including 100% of culture-positive cases, except for 2% with polymicrobial growth [47]. These innovative modalities aid in identifying the exact species involved in the infection and facilitating appropriate treatment.

The internal transcribed spacer (ITS) region is currently believed to be a sequence with great potential for identifying the widest possible range of fungal species, and is nowadays used as a universal DNA barcode for fungal groups [48].

#### 3.5.2. Molecular Diagnostic Techniques Directly Applied to Clinical Samples

Metagenomic deep sequencing (MDS) is nowadays known as a promising approach for better diagnostic sensitivity and accuracy [23]. DNA-sequence-based methods are used for the more rapid species identification of an organism [49]. Lalitha et al. reported their experience with MDS in 46 corneal ulcer cases, evaluating the specificity and sensitivity of traditional methods and DNA and RNA sequencing using latent class analysis (LCA). The sensitivity of MDS was found to be 74%, outperforming KOH/Gram stains (70%) and cultures (52%). On LCA, RNA sequencing demonstrated 100% sensitivity and specificity for bacterial keratitis and 100% sensitivity and 97% specificity for fungal cases [24]. As it is not yet FDA approved, genotyping is performed only in selective cases, but it holds promise in distinguishing a causative pathogen from colonization or contamination [50].

Custom tear proteomic approaches might have an essential role in the future treatment of fungal corneal disease [51]. Genomic approaches, mainly built on distinguishing amplicons of ribosomal RNA genes, are nowadays adopted in clinical practices. The metagenomic approach utilizes 16S rRNA genes to track dynamic transformations in conjunctival flora in mycotic keratitis [52,53]. Diagnostics based on 18S rRNA target enrichment sequencing show potential for diagnosing fungal corneal infections [54].

The PCR technique has the highest positive detection rate overall in cases with culture- or smear-negative results. Molecular characterization can distinguish various species of fungi and can recognize rarer species of fungi, which may pose a problem during diagnosis using only traditional methods. Various molecular methods are used for diagnosing and identifying causative agents in fungal keratitis, including traditional, nested, real-time, multiplex, and conventional PCR followed by enzymatic digestion, sequencing, single-strand conformation polymorphism (SSCP), next-generation sequencing combined with computational analysis dot hybridization, and high-resolution melting analysis [55,56,57,58,59]. A study had shown an important association between culture-proven fungal keratitis and multiplex PCR, reporting 94.1% diagnosis for *Fusarium*, 63.6% for *Aspergillus fumigatus*, and lastly, 100% for *Aspergillus flavus* [55]. The significance of precise fungal detection using molecular diagnostic methods, such as PCR, for optimal management and an improved therapeutic effect has been emphasized [60]. PCR cannot be used to monitor the response of the patient with fungal keratitis to antimycotic therapy because it is cannot differentiate viable from nonviable fungi [23]. 

The rapidity and accuracy of PCR diagnostic methods advocate for their application in the diagnosis of mycotic keratitis. Although not affordable in many clinical centers in developing countries, they should become part of the diagnosis algorithm alongside microscopic evaluation and cultures. MALDI-TOF MS is a novel diagnostic method, which may be reliable and easy to use. It exhibits both high sensitivity and specificity, but has yet to identify and distinguish related fungal species. MDS is a new technique for the diagnosis of fungal keratitis that is able to identify any organism in a single assay. 

## 4. Treatment Approaches

Antifungal agents used in ophthalmology are administered topically, orally, or locally, such as intra-cameral and intra-corneal injections. These agents belong to various drug classes, including polyenes (e.g., amphotericin B and natamycin), triazoles (itraconazole, voriconazole, fluconazole, posaconazole), azoles or imidazoles (e.g., ketoconazole, clotrimazole, econazole, tinidazole and miconazole), and echinocandins (micafungin and caspofungin) [61]. Fungal keratitis usually responds over an extended period of weeks to antifungal therapy, and signs of improvement in an ulcer include a decrease in pain and also in the size of the infiltrate, disappearance of the satellite lesions, rounding out of the feathery margins of the corneal ulcer, and hyperplastic masses in the region of healing fungal lesions. A few advantages and disadvantages to known antifungal therapies are provided in Table 2 [2,62,63,64].

### 4.1. Topical and Systemic Therapy

Polyenes and azoles are essential in the topical treatment of corneal mycotic infections. Natamycin, the only FDA-approved antifungal formulation for ocular fungal infections, is commonly used. The treatment duration for fungal keratitis is often prolonged, with cases requiring weeks to months for complete resolution. Natamycin 5% drops are commonly used for filamentous fungi, while amphotericin B 0.15% is used for yeast-like fungi [6]. Recently developed azoles, such as voriconazole, are more and more utilized due to their broad spectrum and improved ocular penetration profile. Reports show that fungal infections can be resolved with topical treatment in 7.6% of patients, while the other 92.4% require surgical intervention [9]. 

Systemic antifungal agents, including ketoconazole, itraconazole, fluconazole, and voriconazole, are used to overcome the limitations of intermittent dosing with topical medications. The role of oral antifungal therapy in managing keratomycosis remains inconclusive. The MUTT-2 trial, evaluating the effectiveness of oral treatment with voriconazole as additional therapy to topical treatment in severe mycotic keratitis, did not find additional benefits [71]. However, a sub-analysis from the MUTT-2 trial suggested a potential advantage of adding oral voriconazole to culture-positive *Fusarium* keratitis, with a reduced rate of perforation, decreased need for PK, reduced scar size, and improved visual acuity at three months [72]. Systemic antifungals are indicated as an adjunctive treatment in specific cases, such as ulcers > 5 mm in size, involvement of >50% stromal depth, recalcitrant infections, bilateral infections, those associated with scleritis, limbal involvement or endophthalmitis, pediatric cases, post-keratoplasty infections, and cases of impending perforation/perforated ulcers [49,73]. The inconclusive role of oral antifungals in fungal keratitis calls for more randomized control trials in this area for a clearer understanding.

### 4.2. Targeted Drug Delivery

Targeted drug delivery using antifungal agents is a strategic approach that involves the precise delivery of medications to the cornea through intrastromal and intracameral injection, optimizing therapeutic effects while minimizing systemic exposure. Localized approaches, such as topical application, can effectively target superficial fungal infections, delivering high concentrations of antifungal agents directly to the affected area while minimizing systemic side effects. Invasive fungal infections may require systemic delivery via intravenous routes. The advantages of targeted drug delivery include enhanced efficacy at the infection site, reduced systemic toxicity, and improved patient compliance. However, challenges such as developing effective drug carriers, ensuring sustained release, and addressing potential barriers to drug delivery must be overcome. Voriconazole (50–100 µg/0.1 mL) and amphotericin B (5–7.5 µg/0.1 mL) are usually used for this method [2]. 

Resistant keratitis [74], mycotic keratitis complicated with endophthalmitis [75], post-PK and post-photorefractive keratectomy fungal keratitis [76] are indications for this method, using intrastromal, intracameral, and intravitreal injections. Amphotericin B has been the most common drug used in the past; however, due to its higher incidence of ocular and systemic complications, such as corneal toxicity and kidney disease, it is increasingly being replaced by the safer choice, voriconazole [74]. Voriconazole is a second-generation azole with better bioavailability against common fungi, such as *Fusarium* spp. and *Aspergillus* spp., and has shown favorable outcomes in multiple case series when added to the standard treatment regimen [77,78]. 

A randomized control study by Narayana et al., evaluating the effectiveness of adding intrastromal voriconazole 1% to a therapeutic protocol for moderate-to-severe mycotic keratitis, highlighted no noticeable advantage in culture positivity (at the three/seven-day mark), scar dimension, visual impairment (3-month mark), or lower rate of corneal perforation when compared with topical natamycin 5% monotherapy [79]. Additional randomized studies are still needed to verify the optimal regimen in terms of the doses, the intervals of time between them, and the approximate number of injections. Several studies have compared the efficacy of the previously mentioned drugs, with various results, and they failed to describe a model-systematized treatment for fungal keratitis (Table 3).

### 4.3. Nanoparticles

Nanotechnology has been explored in the field of ophthalmology, particularly in the development of novel drug delivery systems (NDDS) and gene delivery. Various types of nanoparticles, such as nanosuspensions, liposomes, nanofibers, and nanotubes, are being investigated for delivering antifungal agents. The aim is to achieve enhanced ocular penetration, retention, and improved bioavailability. Sushma et al. have synthesized and characterized ethosomes (IAEs) encapsulating a dye, indocyanine green, and an antifungal drug represented by amphotericin B, in order to achieve combinational photothermal therapy for fungal keratitis. This nano-formulation exhibited a synergistic and sustained antifungal effect, giving way to more in-depth studies [90]. Kumar et al. have developed nanostructured lipid carriers loaded with itraconazole and shown that this system can be an effective approach, with adequate activity against fungi and less local side effects [91]. While several in vitro and in vivo experimental studies have highlighted promising results using this treatment method, further controlled studies are necessary for its use on humans [65,66,67,68,69,70,71,72,73,74,75,76,77,78,79,80,81,82,83,84,85,86,87,88,89,90,91,92,93].

### 4.4. Therapeutic Contact Lenses

A therapeutic contact lens represents an ideal drug delivery system for the continuous provision of medication to the affected cornea, simultaneously limiting nonspecific absorption and drug loss through tears. One of the benefits of these inserts is having a higher precorneal residence time, which provides the means to release the drug material at a preprogrammed rate and leads to higher bioavailability and prolonged drug activity. The post-lens tear film, formed behind the contact lens, has reduced tear mixing and exchange; therefore, drugs released from the contact lens into this tear film will have a prolonged contact time with the cornea [91]. Although some studies in this field have shown promise, there is currently no FDA-approved therapeutic contact lens for mycotic keratitis [94].

### 4.5. Photodynamic Therapy

Photodynamic therapy (PDT) is a treatment method that utilizes a photosensitizing agent, which, when activated by specific wavelengths of light, produces reactive oxygen species. These reactive oxygen species can cause cellular damage, leading to the destruction of targeted cells. In the context of corneal infections, including *Acanthamoeba* and fungal keratitis, PDT has been considered as an alternative treatment modality [95]. Photosensitizing agents, such as methylene blue or rose bengal, are applied to the corneal lesion. Subsequent exposure to light, typically using a laser of a specific wavelength, activates the photosensitizer, generating reactive oxygen species that can selectively damage fungal cells. More important, PDT can destroy fungi nonselectively. PDT-associated genotoxic or mutagenic effects on fungal or human cells have so far not been observed [90]. The application of PDT for fungal keratitis aims to achieve localized and targeted antifungal effects while minimizing damage to healthy tissue. It is often considered in cases where conventional antifungal therapies may be insufficient or in situations where surgical intervention is challenging. PDT offers advantages such as negligible drug resistance, high spatiotemporal control, and fewer side effects. 

The PDT regimen for corneal infections resembles the treatment used in keratoconus cases, involving the use of photosensitizers and UV-A light. Experimental studies comparing different photosensitizers have shown promising results. For example, a study has compared rose bengal with riboflavin PDT and has indicated that rose bengal PDT with green light showed greater effectiveness in vitro against common fungi, such as *Fusarium solani*, *Aspergillus fumigatus*, and *Candida albicans* [96]. Other novel treatments studies confirm that the UPR (fungal unfolded protein response) is essential for Aspergillus fumigatus to establish infection in the cornea, and its inhibition with the Ire1 inhibitor, 4μ8C, can significantly reduce fungal growth in mice [97]. Even though antimicrobial PDT has shown potential in treating several therapy-refractory diseases in vitro and in animal studies, there are no large-sized clinical studies currently available.

### 4.6. Corneal Crosslinking

Corneal collagen crosslinking (PACK-CXL) is a well-established procedure commonly used for managing ectatic corneal disorders, as well as conditions like bullous keratopathy. The procedure uses the ultraviolet-A irradiation of the cornea that has been primed with the photosensitizer riboflavin (vitamin B2). This process leads to the formation of reactive oxygen species and singlet oxygen, ultimately increasing the corneal biomechanical stability by forming covalent bonds between stromal collagen fibrils [98]. The CXL method can be involved in three main mechanisms in the pathophysiology of fungal keratitis: antimicrobial activity, anti-inflammatory action, and a higher resistance of the cornea to enzymatic degradation [96].

Several experimental studies have indicated that corneal collagen crosslinking can be an effective adjunctive treatment for fungal keratitis, especially when combined with antifungal agents early in the disease course [99,100]. A few studies have shown that extending the duration of irradiation and elevating the concentration of riboflavin may enhance the effectiveness of this method [99,101]. While some case reports and studies have shown it to be a useful adjunctive therapy, the Cross-Linking Assisted Infection Reduction Trial (CLAIR trial), a randomized controlled study, concluded that corneal collagen crosslinking has no added advantage and may lead to higher vision impairment compared to standard treatment in mycotic keratitis [102,103,104].

### 4.7. Surgical Approach

Surgical treatment plays a crucial role in the management of fungal keratitis, with approximately 50% of cases requiring therapeutic PK to control the infection [71]. The incidence of PK in fungal keratitis varies depending on factors such as the geographical prevalence of fungal infections, local healthcare practices, and the efficacy of early antifungal interventions. In a recent study, 50% of the cases with mycotic keratitis needed PK [40].

The common indications for PK in fungal keratitis include perforated ulcers, impending perforations, and cases that do not respond to conservative management. The necessity for PK in fungal keratitis varies between 15% and 55%, underscoring the fact that sole reliance on medical treatment may not invariably lead to success [105,106]. A secondary analysis of data from the MUTT-2 trial identified several risk predictors that may indicate further need of a PK—infiltrate dimensions and depth, and hypopyon in the anterior chamber—predictors that could indicate the need for PK in fungal keratitis [107]. For example, the presence of hypopyon increased the odds, and an increase in infiltrate size indicated a higher likelihood of requiring PK.

PK poses several significant complications, such as graft rejection, microbial/fungal infection, or secondary glaucoma. The rate of re-infection post keratoplasty is of utmost concern in mycotic keratitis, between 6% and 16% [94,100,104].

Additionally, the delayed use of steroid drops post keratoplasty, often due to the fear of re-infection, can lead to increased inflammation, graft decompensation, and vascularization, contributing to poorer surgical outcomes. Overall, surgical intervention, particularly PK, remains an important aspect of the comprehensive management of fungal keratitis, but challenges in terms of donor availability and postoperative care need to be addressed for better outcomes [94]. Cyanoacrylate tissue adhesive and bandage contact lens may also be used in the management of microperforation or impending perforation.

## 5. Conclusions

Fungal keratitis is common in warm and humid regions, with more than 100 fungal species that can cause sight-threatening keratitis. The etiopathogenesis involves morphological changes, trauma, adhesion, virulence factors, and immune response. In several cases, the traditional diagnostic approaches, such as smears and cultures, have failed to provide reliable diagnosis, leading to refractory fungal keratitis and poor prognosis. In contrast to conventional diagnostic methods, new methods based on molecular biology, such as PCR, DNA and RNA sequencing, and IVCM, can improve the diagnosis of fungal keratitis and optimize the treatment to obtain a better visual acuity and a smaller scar. There have also been improvements in treatment, with several different approaches to mycotic keratitis attempted, especially in refractory cases. Newer antifungal agents and combination treatment, in comparison to monotherapy, have been shown to be more effective in the management of mycotic keratitis. With the progress that has been made in the pathogenesis and diagnosis of fungal keratitis, more treatment strategies will undoubtedly be developed to reduce the socioeconomic burden related to fungal keratitis.

## Figures and Tables

**Table 1 microorganisms-12-00161-t001:** Brief description of traditional and novel diagnostic methods.

Diagnostic Method	Definition	Advantages	Disadvantages	Year of Implementation
Microscopic exam [1,2,10]	Direct visualization of fungal elements in corneal scrapings using different staining methods	Rapid results, low cost	Limited sensitivity, expertise required	19th century
Culture [2,21,22]	Growth of fungal organisms on specific culture media	Definitive identification of fungal species	Slow turnaround time, requirement for specialized laboratory facilities	20th century (Sabouraud/agar introduced)
Polymerase chain reaction (PCR) [2]	Amplification of fungal DNA in a sample to identify species	High sensitivity and specificity, rapid results	Technical complexity, need for trained personnel	1983 (for PCR)
In vivo confocal microscopy (IVCM) [2]	Real-time imaging of corneal structures and fungal hyphae	Non-invasive, high resolution, early detection of fungal elements	Equipment cost, limited availability, operator expertise required	1980s
Antifungal susceptibility testing (AFST) [1,25]	Tests the sensitivity of fungal isolates to various antifungal drugs	Guides treatment decisions, tailored therapy	Time-consuming, resource-intensive	1970s
Metagenomic deep sequencing (MDS) [23,24]	Comprehensive genomic analysis of all genetic material in a sample	Identifies diverse pathogens, including fungi	High cost, complexity in data interpretation	2000s
Matrix-assisted laser desorption ionization time-of-flight mass spectrometry (MALDI-TOF MS) [2]	Identification of microorganisms through mass spectrometry	Rapid identification of microorganisms, high accuracy	Equipment cost and maintenance, limited database coverage	1990s

**Table 2 microorganisms-12-00161-t002:** Antifungal agents in fungal keratitis.

Treatment	Indication	Administration Route	Benefits	Disadvantages
Natamycin [22,62]	Wide spectrum of activity (*Fusarium* spp., *Aspergillus* spp., and others)	Topical	Commercially available, better clinical and microbiological outcomes, lower rate of corneal perforation (MUTT1)	Poor ocular penetration, long treatment and increased expenses, effective only when applied topically
Amphotericin B [2,62]	*Candida* spp., *Aspergillus* spp., and *Cryptococcus*	Topical, intravenous, intrastromal, intravitreal	Penetrates the deep corneal stroma after topical application, good bioavailability after topical use	Many side effects (systemic and subconjunctival administration), poor ocular penetration after systemic use, not commercially available
Voriconazole [2,62,63]	*Aspergillus* spp., *Candida* spp., *Fusarium* spp., and *Cryptococcus* spp.	Topical, oral, intrastromal, intracameral, intravitreal	Alternative drug for recalcitrant cases, adjunctive in severe fungal keratitis, most common intrastromal antifungal agent, oral treatment has high bioavailability and can penetrate several parts of the eye	Poorer outcomes for treatment of *Fusarium* (MUTT-1)
Miconazole [6,62]	*Scedosporium apiospermum*	Topical, subconjunctival, intravenous	Topical and subconjunctival administration generally well tolerated	Systemic administration can lead to toxicity
Itraconazole [65]	*Candida* spp., *Aspergillus* spp.	Topical, oral	Nanosized carriers lead to better absorption	Side effects, poor ocular bioavailability
Posaconazole [66,67]	*Aspergillus* spp., *Fusarium* spp. *Candida* spp., and rare fungi	Topical, oral	Low maximum inhibitory concentration (MIC) values, minor side effects	
Ketoconazole [62,68]	*Candida* spp. and other molds	Topical	Well absorbed and good tissue distribution after oral administration	Hepatotoxic side effects
Luliconazole [64,69]	*Fusarium* spp. and other common fungi	Topical, nanoemulsion formulation (not yet included in clinical trials)	Broad spectrum of activity, lower MIC	Higher MIC against *Candida* spp.
Echinocandins [2,49,70]	*Candida* spp., *Aspergillus* spp., less effective on *Fusarium* spp.	Topical, oral	Reduced toxic side effects	Usually used for systemic fungal infections

**Table 3 microorganisms-12-00161-t003:** Recent studies on antifungal agents and their clinical responses and outcomes.

Study	Route of Administration	Material	Results
Voriconazole 1% vs. natamycin 5% [80]	Topical	120 patients	No significant difference (VA, scar size, or complications)
Voriconazole 1% vs. natamycin 5% [81]	Topical	30 patients	All patients with natamycin healed; 14/15 patients with voriconazole healed
Voriconazole 1% vs. amphotericin B 0.3% vs. fluconazole 0.2% [82]	Topical	44 rabbits (animal studies)	No significant difference in healing, low toxicity for voriconazole
Voriconazole vs. amphotericin B (several dosages) [83]	Topical	Colony counts	Amphotericin led to lower colony counts
Voriconazole (50 µg/0.1 mL) vs. amphotericin B (5 µg/0.1 mL) vs. natamycin (10 µg/0.1 mL) [84]	Intrastromal	60 patients	No significant difference (VA)—natamycin showed faster healing; amphotericin B had a higher rate of deep vascularization after healing
Voriconazole 1% vs. natamycin 5% [85]	Topical	323 patients	Natamycin was associated with significantly better clinical and microbiological outcomes
Econazole 2% vs. natamycin 5% [86]	Topical	112 patients	No significant difference
Itraconazole 1% vs. natamycin 5% [87]	Topical	100 patients	Natamycin was associated with a better clinical response
Natamycin 5% vs. amphotericin B 0.15% vs. voriconazole 1% vs. and fluconazole 0.2% [88]	Topical	Colony counts	Amphotericin B and natamycin had equal effectiveness and full inhibition
Micafungin 150 µg vs. natamycin 5% [89]	Topical	18 rabbits (animal studies)	No significant difference (scar size, infiltrate)

## Data Availability

No new data generated.

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
