# Peer review of "Recent Advances in Diagnosis and Treatment Approaches in Fungal Keratitis: A Narrative Review"

_microorganisms, 2024, doi:10.3390/microorganisms12010161_

Round 1
Reviewer 1 Report
Comments and Suggestions for Authors
This article comprehensively reviewed the recent advances in diagnosis and treatment of fungal keratitis. Some comments are listed below.
In general, the scientific names (genera and species) of microorganisms should be italicized.
2.2. Culture growth: Calcofluor-white is a fluorescent blue dye, not a culture media. Please check.
2.4 Antifungal susceptibility testing:
1) LN 191-194: “Two universally recognized standard method bodies, the CLSI and EUCAST, have established phenotypic assays for in vitro AFST based on the broth micro-dilution method for both filamentous fungi and yeasts [29,30].”→ Only references for yeast AFST have been provided here. Please also provide references for AFST for filamentous fungi.
2) LN 196-198: “CLSI has provided guidelines on MIC breakpoint values for Candida species, while EUCAST has set breakpoints for various antifungal agents against specific Aspergillus species.” → EUCAST also provides MIC breakpoints for yeasts. Please check the EUCAST website.
2.5. Molecular diagnostic techniques:
1) Please separate the issue into two parts: (1) identification of microorganisms (isolates obtained from culture), and (2) molecular identification of fungal pathogens directly on clinical samples (cornea scrapings…). Molecular methods for two intentions are different.
2) “Internal transcribed spacer (ITS) sequencing” is one of the most used DNA sequence-based methods for identification of fungi. May consider mentioning ITS sequencing in the text.
3) LN 211-246 “While not yet FDA- approved, genotyping is performed only in selective cases, but it holds promise in distinguishing a causative pathogen from colonization or contamination.” → It is interesting that genotyping can differentiate between infection and colonization/contamination. Please cite references here.
Table 1:
1) Please provide references for each drug as noted in Table 2 if possible.
2) Please consider adding a column to indicate the administration routes of each drug: intravenous, topical, oral, etc. ?
3) Amphotericin B: good bioavailability mentioned in benefit but poor ocular penetration mentioned in disadvantage? They seem to be conflicting. Please also briefly describe bioavailability here: transcorneal bioavailability?
Table 2: Colony count is not “number of patients”. Please revise the subtitle of this column.
3.3. Nanoparticles: The authors may consider adding itraconazole nanoparticles as examples in this paragraph, which have been mentioned in Table 1 and references.
4. Conclusions: LN 446: ARN sequencing, type error
Author Response
Dear reviewer, thank you so much for taking the time to evaluate our manuscript and for all your comments. I will further address each one of them:
- We italicized all the names and species
- LN 191-194: “Two universally recognized standard method bodies, the CLSI and EUCAST, have established phenotypic assays for in vitro AFST based on the broth micro-dilution method for both filamentous fungi and yeasts [29,30].”→ Only references for yeast AFST have been provided here. Please also provide references for AFST for filamentous fungi.We provided a separate bibliography also for filamentous fungi
- LN 196-198: “CLSI has provided guidelines on MIC breakpoint values for Candida species, while EUCAST has set breakpoints for various antifungal agents against specific Aspergillus species.” → EUCAST also provides MIC breakpoints for yeasts. Please check the EUCAST website.We also reviewed MIC values for yeasts, thank you very much
- Please separate the issue into two parts: (1) identification of microorganisms (isolates obtained from culture), and (2) molecular identification of fungal pathogens directly on clinical samples (cornea scrapings…). Molecular methods for two intentions are different.We also separated this part as requested, we categorized each method into one of these two subsections, based on which is the most commonly used (in cases where both scrapings and cultures can be used)
- “Internal transcribed spacer (ITS) sequencing” is one of the most used DNA sequence-based methods for identification of fungi. May consider mentioning ITS sequencing in the text.We also added a small section about ITS sequencing
- LN 211-246 “While not yet FDA- approved, genotyping is performed only in selective cases, but it holds promise in distinguishing a causative pathogen from colonization or contamination.” → It is interesting that genotyping can differentiate between infection and colonization/contamination. Please cite references here.This was referring to differentiation in pulmonary diseases, where it was found that genotyping might show this advantage. From our knowledge, in ocular diseases this kind of study hasn't been done
- Table 1: we added references for each drug mentioned, we added a column stating which drug has multiple routes of administration and which are they and we also made it clear regarding bioavailability vs poor ocular penetration (in poor ocular penetration we were referring to systemic treatment)
- Table 2: Colony count is not “number of patients”. Please revise the subtitle of this column. We revised the column title
- Nanoparticles: The authors may consider adding itraconazole nanoparticles as examples in this paragraph, which have been mentioned in Table 1 and references.We added information about itraconazole nanoparticoles
- Conclusions: LN 446: ARN sequencing, type error . We are so sorry about this, ARN is the term for RNA in our language, we fixed itThank you once again for your time. Assist prof Dr Ghenciu Laura Andreea
Reviewer 2 Report
Comments and Suggestions for Authors
Comments to the Authors
This review titled “Recent advances in diagnosis and treatment approach in fungal
keratitis: a narrative review.The authors mainly reviewed novel diagnostic methods include PCR-based approaches, metagenomic deep-sequences, in vivo confocal microscopy and antifungal susceptibility-testing. The language and arrangement of article is not good. Thus, this article can be accepted after major revison.
Main conments
1. The author should check the sentence tense and grammar.
2. Although authors reviewed the novel diagnostic methods of fungal
keratitis, the describtion of various kinds of fungal should be added.
3. The author should summarize the Table for novel diagnostic methods in Section 2. It is ok if authors can cite the latest diagnostic figures.
4. The discussion is poor, the detailed summary and discussion is very key for a excellent Review.The authors shoud further sum up the the perspective of each method after each paragraph.
Thus, this article can be accepted after major revison.
Comments on the Quality of English LanguageModerate editing of English language required
Author Response
Dear reviewer, thank you so much for taking the time to evaluate our manuscript and for all your comments. I will further address each one of them:
1.Although authors reviewed the novel diagnostic methods of fungal
keratitis, the describtion of various kinds of fungal should be added.
We added a section about the most important types of fungi that can cause fungal keratitis, stating clinical and environmental implications
2. The author should summarize the Table for novel diagnostic methods in Section 2. It is ok if authors can cite the latest diagnostic figures.
In the beginning of the third section we added a table including the most important methods, the year since it has been used, advantages and disadvantages
3. The discussion is poor, the detailed summary and discussion is very key for a excellent Review.The authors shoud further sum up the the perspective of each method after each paragraph.
We also added a small paragraph to summarize each method described in the article
We had a colleague fluent in English check the grammar and topic and modified the manuscript where necessary.
Thank you once again for your time.
Assist prof Dr Ghenciu Laura Andreea
Round 2
Reviewer 2 Report
Comments and Suggestions for Authors
Accept